# Modeling the Design Characteristics of Woven Textile Electrodes for long−Term ECG Monitoring

**DOI:** 10.3390/s23020598

**Published:** 2023-01-04

**Authors:** Peter J. Brehm, Allison P. Anderson

**Affiliations:** Aerospace Engineering Department, University of Colorado, Boulder, CO 80309, USA

**Keywords:** circuit model impedance, conductive textile manufacturing, electrocardiograph, MATLAB, transfer function, skin−electrode interface model, woven electrode

## Abstract

An electrocardiograph records the periodic voltage generated by the heart over time. There is growing demand to continuously monitor the ECG for proactive health care and human performance optimization. To meet this demand, new conductive textile electrodes are being developed which offer an attractive alternative to adhesive gel electrodes but they come with their own challenges. The key challenge with textile electrodes is that the relationship between the manufacturing parameters and the ECG measurement is not well understood, making design an iterative process without the ability to prospectively develop woven electrodes with optimized performance. Here we address this challenge by applying the traditional skin−electrode interface circuit model to woven electrodes by constructing a parameterized model of the ECG system. Then the unknown parameters of the system are solved for with an iterative MATLAB optimizer using measured data captured with the woven electrodes. The results of this novel analysis confirm that yarn conductivity and total conductive area reduce skin electrode impedance. The results also indicate that electrode skin pressure and moisture require further investigation. By closing this gap in development, textile electrodes can be better designed and manufactured to meet the demands of long−term ECG capture.

## 1. Introduction

Wellness Monitoring is the concept of using wearable devices coupled with automated data analysis to monitor individualized health status passively and continuously over time [1]. One category of wellness monitoring is the continuous capture of electrocardiograph (ECG) data. The ECG is a graph depicting the electrical impulses generated by the heart during its cardiac cycle. According to the Center for Disease control (CDC), cardiovascular disease (CVD) has been a leading cause of death for decades [2,3]. By enabling proactive intervention through wellness monitoring, many of the costly or detrimental reactive treatments of CVD could be avoided [4,5]. Beyond tracking life threatening diseases, wellness monitoring also has the potential to help people such as first responders and athletes through performance optimization [6,7]. The body’s heart rate is impacted by exercise, sleep, stress, emotion, and many other stimuli [7,8,9,10,11]. Wearable devices have the potential to improve quality of life by providing real time feedback to the wearer about their own biological status [12].

The main challenge with long−term ECG monitoring is that the adhesive gel electrodes commonly used with ECG capture were not optimized for long duration applications [4,13]. These clinical adhesive sensors consist of a silver-silver chloride (Ag/AgCl) metal electrode, a gel electrolyte, and a skin adhering patch [14]. In the context of long term wellness monitoring their limitations are: they are designed to be single use, they must be applied fresh on specific body locations, the wet electrolyte gel dries over time, and the adhesive irritates the skin with repeated application and removal [14,15,16,17,18]. Additionally, every re-application increases the risk of placement error which reduces the diagnostic capabilities of the measurement [19].

To address the need for long−term ECG monitoring alternative electrode sensors have been, and are being, developed which do not have the same limitations [4,20]. In this work, we focus on conductive woven textile electrodes. These sensors are created on a loom of interlaced perpendicular conductive and non-conductive yarns [21]. Woven electrodes are reusable, in-extensible, and can be integrated directly into a garment enabling consistent placement and increased comfort [15]. They do not use a gel electrolyte, meaning performance does not degrade over time. Additionally, they do not use adhesive to adhere to the body, ensuring they do not irritate the skin on removal and reapplication [22]. These benefits, though, are also their primary challenge. Without gel electrolyte they have a comparatively higher skin−electrode impedance, and without adhesive they are comparatively more susceptible to motion related noise. Thus, there is a desire to design these types of sensors optimally to increase signal throughput, and therefore increase signal robustness.

While woven electrodes have appealing properties for long duration applications, there is a limited understanding of how to best optimize their electrical performance to overcome their weaknesses. Previously, the design of textile electrodes has been evaluated through iterative fabrication and testing cycles. Since these woven electrodes are a relatively new technology there is a gap in the design of woven electrodes between the manufacturing parameters and their electrical performance [23].

This paper performs an analysis of a comprehensive sweep of novel woven electrode types to provide a novel understanding of which manufacturing design parameters impact the electrical characteristics of the electrode. The traditional skin−electrode interface model contains circuit elements describing the electrical behavior of an ECG wave as it travels from the heart to the sensing circuitry. By applying this model to a variety of woven electrodes and their corresponding measured data sets it is possible to show how their manufacturing parameters impact the measured electrical signal. Specifically, we show how yarn type, pattern type, and surface area are related to the skin−electrode impedance and correspondingly the ECG waveform. This relationship provides new insight into the design of textile electrodes and informs new areas of exploration for even better designs.

## 2. Materials and Methods

In order to investigate the parameter space of yarn type, weave pattern and surface area, electrodes manufactured by Arquilla et al. were used for analysis [8,15,21]. Seventeen distinct types of electrodes are considered: 8 in a pattern and yarn type sweep, 8 in an area sweep, and 1 adhesive gel electrode type. The weave patterns are shown in Figure 1 and the surface area sweep is shown in Figure 2. The yarn types are silver coated nylon thread with resistivity of 42.65 Ohms/m and spun steel thread with a resistivity of 91.86 Ohms/m.

Table 1 shows the full the combination of electrode types and their corresponding manufacturing parameters used for data collection and analysis.

For each electrode type, approximately 2 min of ECG data was collected from 6 seated human subjects using the standard 3 lead placement method. Data was collected using the BIOPAC MP160 circuitry with the wireless BioNomadix BN-ECG2 transceiver. The transceiver has a front-end band pass filter with cutoff frequencies at 1Hz and 35 Hz. There is also a 60 Hz narrow band notch filter and a 66 dB voltage amplifier with a 2 MΩ differential input impedance [8,25]. The full description of the data collection procedure is outlined by Arquilla et al. [21].

Simplifying assumptions were made to focus the scope of analysis on the circuit parameters of the woven electrodes. First, constant values for skin properties are used across human subject. It is outside the scope of the analysis to see how skin variability impacts ECG measurement. Therefore constant terms for skin are used in the skin−electrode interface model. Another assumption is that consistent application was used during data collection. It is outside the scope to evaluate how location and pressure of the electrode on the skin impact the measurement. Next, idealized BIOPAC circuit components are assumed for modeling. The filters are ideal 2nd order filters. The Amplifier provides flat gain across frequency and there is negligible frequency-dependent loss in the BIOPAC circuitry. Last the “in−body” waveform for each subject is constant between data sets. This assumption means the electrode does not influence the signal generated by the body, only the signal measured by the circuitry.

### 2.1. Procedure

A simplified block diagram of the ECG system is represented in Figure 3, which relates the “in−body” ECG waveform to the measured output waveform through a parameterized transfer function. The transfer function represents the generalized ECG hardware from the surface of the skin to the measured signal output. It consists of the traditional skin−electrode interface model, and the circuit model of the front end ECG circuitry. Both of these models are described in the model subsection.

The general simulation method relies on the ability of the transfer function to transition between arbitrary input and output waveforms. Fundamentally the transfer function is an equation with three unknowns: the input, the output and the function relating them. If two are known then the third can be solved. This characteristic is used repeatedly across data sets to identify the unknown circuit parameters of the woven electrode transfer function. The generalized equation for the transfer function is shown in Equation (Equation 1). Here the ratio of input and output voltage waveforms is related by a ratio of polynomials with coefficients. FFT denotes the Fast Fourier Transform converting time domain to frequency. Each $A_{i}\left(i=0,1,\ldots n\right)$ and $B_{i}\left(i=0,1,\ldots m\right)$ coefficient is determined by the circuit parameters, and ultimately the manufacturing parameters, of the particular electrode type.
(1)$$\frac{FFT\{V_{out}\left(t\right)\}}{FFT\{V_{in}\left(t\right)\}}=\frac{V_{out}\left(s\right)}{V_{in}\left(s\right)}=TF\left(s,A_{0,1,\ldots ,n},B_{0,1,\ldots ,m}\right)=\frac{A_{n}s^{n}+\ldots A_{2}s^{2}+A_{1}s^{1}+A_{0}}{B_{m}s^{m}+\ldots B_{2}s^{2}+B_{1}s^{1}+B_{0}}$$

The analysis process begins with simplifying measured ECG signal data. In order to exclusively observe the effect of the electrodes, the noise artifacts and human subject variations are removed. The mean ECG waveform was calculated for each data set. In this context the “mean ECG waveform” is defined as the average PQRST waveform produced during the 2 min collection window. The mean waveform was calculated by identifying R-peaks, rejecting outlier peaks, then aligning them in time and calculating the mean. Peaks were identified using the MATLAB findpeaks() function, and outliers were identified using MATLAB’s isoutlier() function. To ensure high quality waveforms, the signal-to-noise ratio (SNR) was calculated for each dataset with MATLAB’s snr() function. Any dataset with SNR < 0 dB was removed. Any dataset with fewer than 30 detected R-peaks was also removed. An example adhesive dataset and the corresponding mean ECG waveform calculated from aligning all the R-peaks is shown in Figure 4. An example ECG dataset captured on the same subject with textile electrodes is also shown in Figure 4. Notice the different Y−axis. The mean waveform is computed for all 17 electrode types for all subjects yielding 82 total waveforms after screening criteria were satisfied.

The adhesive ECG dataset is the “gold standard” since it was captured with well documented clinical adhesive gel electrodes. Furthermore all the coefficient parameters in the transfer function polynomial are known for the adhesive electrode data sets [24]. Using the mean ECG waveform and the know transfer function of Adhesive electrodes, the in−body signal is calculated using Equation (Equation 1) for each subject. Specifically, the input waveform, $V_{in}\left(s\right)$, equals the measured output waveform ,$V_{out}\left(s\right)$, multiplied by the inverse transfer function, $TF^{-1}\left(s\right)$. We take this in−body signal to be the “true” signal emitted from the heart to be used in the next step in the process.

The next step of the process is to solve for the unknown parameters of the transfer function for the woven electrodes. This is accomplished by using the mean output waveform of the woven electrodes and the mean input waveform per subject that was just computed. Since the transfer function itself consists of multiple parameters, the transfer function coefficients cannot be calculated directly. A parametric optimizer is used to fit the coefficients by minimizing a cost function. The cost function will be discussed more in the model section.

These above steps are repeated for each woven electrode type forming a complete set of electrical transfer function parameters per electrode type. The last step is to validate the model’s ability to simulate waveforms across human subjects. A Leave−One−Out Cross−Validation procedure was performed. Once all these steps are completed the results are compared to search for trends between electrical parameters and manufacturing parameters.

### 2.2. Models

All models were developed in MATLAB 2021. The first model is the traditional skin−electrode interface model. The ECG voltage originates at the heart and travels outwards to the skin where it is transduced from flowing ions to electrical current by the electrode. The interface is modeled electrically by two stages of parallel R/C pairs connected by series resistors. For the purpose of the equivalent model, the in−body signal does not need to be replicated in ionic current units, so the voltage dependent voltage sources have been excluded from the model. The circuit model and skin cross section are shown in Figure 5.

In this model: $R_{u}$ represents the resistance of the dermis, and subcutaneous layers, $R_{e}$ is the resistance of the epidermis, $C_{e}$ is the capacitance induced by the stratum corneum, $R_{s}$ is the resistance of the electrolyte, $R_{d}$ is the charge transfer resistance in the electrode, $C_{d}$ is the capacitance across the electrode-electrolyte interface, and $R_{Lead}$ is the resistance of the lead wire to the input circuitry. In the case of textile electrodes, initially there is no electrolyte. However, after a few minutes minuscule amounts of perspiration accumulate and behave the same as a thin layer of electrolyte, meaning from a modeling perspective they have the same circuit structure as wet electrodes when used for wellness monitoring [12,18,26]. The cumulative impedance of these circuit elements across the interface, defined as $Z_{E}$, is expressed in the Laplace domain in Equation (Equation 2). The subscript *E* denotes the particular electrode type under consideration.
(2)$$Z_{E}\left(s\right)=R_{Lead}+\frac{R_{d}}{1+sR_{d}C_{d}}+R_{s}+\frac{R_{e}}{1+sR_{e}C_{e}}+R_{u}$$

From the electrodes, the lead wires connect to the input of the BIOPAC system. The BIOPAC circuitry is a differential OpAmp cascaded by a band pass filter (BPF), and notch filter. The complete ECG system model is therefore two electrode−interfaces connected to the differential inputs of the amplifier followed by the filters. The schematic representation for the system is shown below in Figure 6.

The cumulative transfer function of the full ECG system in terms of knowns and unknowns is shown in Equation (Equation 3). This is found from Equation (Equation 1) by expressing each block in transfer function notation as a ratio of output to input voltages. The amplifier’s input impedance forms a voltage divider circuit with the skin electrode impedances. This is then cascaded with the amplifier gain, *G*, then the transfer functions of the two filters, $TF_{BPF}$ and $TF_{Notch}$. In this equation $R_{amp}$ is the high input impedance of the amplifier, and $Z_{E}$ is the skin electrode impedance from Equation (Equation 2). After plugging all the terms the equation is expressed as the ratio of two 2nd order polynomials.
(3)$$TF_{E}\left(kno,ukn\right)=\frac{R_{amp}/2}{Z_{E}+R_{amp}/2}*G*TF_{BPF}*TF_{Notch}=\frac{A_{2}s^{2}+A_{1}s^{1}+A_{0}}{B_{2}s^{2}+B_{1}s^{1}+B_{0}}$$

The coefficients of the polynomial ratio depend on the elements in Equation (Equation 2) as well as the circuit parameters of the BIOPAC circuitry. Known terms are the opamp gain, filter cutoff frequencies and widths, skin structure parameters $C_{e}$, $R_{e}$, and $R_{u}$ and the lead wire resistance $R_{Lead}$. The unknown terms are the electrodes specific parameters, $C_{d}$, $R_{d}$, and $R_{s}$. This means that the coefficients $A_{0,1,2}$ and $B_{0,1,2}$, are also functions of the known and unknown terms. The three unknown terms are what we are specific trying to identify for each of the woven electrodes.

The unknown parameters are found by using a MATLAB cost function optimizer, fmincon(). The optimizer works by finding a constrained minimum of a provided multi−variable cost function. Specifically, the “in−body” signal, calculated from the Adhesive electrode data, is propagated through a transfer function to produce a simulated ECG waveform. This is described in Equation (Equation 4), which is just a rephrasing of Equation (Equation 1).
(4)$$V_{Simulated}\left(s\right)=TF_{E}\left(kno,ukn\right)*V_{Measured}\left(s\right)$$

The cost function in Equation (Equation 5) is calculated as the sum of the squared differences which quantifies the difference between the simulated and measured voltage waveforms. The unknown values of the transfer function, $C_{d}$, $R_{d}$, and $R_{s}$, are iteratively generated from an initial bounded guess by the optimizer until the best fit values are found. The parameters, which produced the minimum cost, are the best fit for the dataset and correspondingly for the electrode type. Since the electrodes are the same between subjects, the transfer function parameters must also be the same. In order to fit the electrode, and not the subject, the cumulative costs of all subjects added together was minimized thereby obtaining a set of parameters which represent the best fit across subjects. Equation (Equation 5) depicts the normalized cost function of all 6 subjects’ cumulative error. The optimizer identifies transfer function parameters which produce the $V_{Simulated}$ waveform. The simulated waveform that produces the lowest normalized cost corresponds to parameters which are the best fit for the particular electrode under test.
(5)$$NormalizedCost=\sum_{AllSubjects}^{}{\left|\frac{V_{Measured}-V_{Simulated}}{max(V_{measured})}\right|}^{2}$$

### 2.3. Leave-One-Out Cross-Validation

The goal of LOOCV is to evaluate how well the predictions made by the model match the observed data. The basic process is to sequentially exclude each subject during the fit optimization step. Then to use the fit parameters to simulate an ECG waveform for the subject which was excluded. The Root mean squared Error (RMSE) shown in Equation (Equation 6) between measured and simulated waveforms is calculated. Here *n* is the number of samples in the average ECG waveform. After every subject has been excluded once and their RMSE is calculated, the average of the RMSE’s is computed for each electrode type.
(6)$$RMSE=\sqrt{\frac{\sum_{n}^{}{\left(V_{Measured}-V_{Simulated}\right)}^{2}}{n}}$$

## 3. Results

The circuit parameters, $C_{d}$, $R_{d}$, and $R_{s}$, were fitted for the 16 woven electrode types. These circuit parameters are presented in a scatter plot versus the corresponding manufacturing parameters of yarn type, weave pattern, and surface area. In addition to the circuit parameters, the skin−electrode impedance $Z_{E}$ is plotted against the manufacturing parameters. Since $Z_{E}$ is frequency dependent, an in−band frequency of 25 Hz is used for display purposes. All fit circuit parameters and their corresponding impedance in magnitude and phase is expressed in Table A1 in the Appendix A. The first manufacturing design parameter comparison in Figure 7 shows the surface area swept across circuit parameters while yarn type and pattern type are constant. Next, Figure 8 shows the surface area swept across skin−electrode impedance while yarn and pattern are constant.

Since pattern and yarn are categorical data types, they are plotted together for simplicity and are all the same surface area. Figure 9 shows the pattern and yarn types versus the circuit parameters, and Figure 10 shows the pattern and yarn types versus skin−electrode impedance. Recall the label column from Table 1. Those labels correspond to the legend labels in the pattern and yarn swept plots below.

After identifying the fit circuit parameters, the simulated mean waveforms for all subjects were plotted and compared against the corresponding measured waveform. The top six best fit waveforms are shown in Figure 11 and the six worst fit waveforms are shown in Figure 12. In this context “best” and “worst” is determined by the sum of squared differences (SSD) between measured and simulated waveforms. Note the different Y−axis of each waveform. The amplitude is a product of each subject and the corresponding electrode used for collection.

## 4. Discussion

The primary contribution of this research is the relation between manufacturing parameters and circuit parameters for this specific set of woven electrodes. This relationship comes from the equivalent model of the ECG system coupled with a comprehensive measured dataset. The performance of the model was confirmed with the Leave−One−Out Cross−Validation test.

While this model is built from the traditional skin−electrode interface model, the specific architecture chosen to model this behavior has implications for the fit parameters. The three fit parameters form a parallel resistor/capacitor (RC) pair in series with another resistor. The series resistance, $R_{s}$, seemed to strongly dominate the impedance term because all frequency content must travel through the series resistance. The parallel RC allows higher frequency content to bypass the resistance $R_{d}$ in favor of the lower impedance $C_{d}$. The $R_{s}$ term traditionally applies to the electrolyte gel, but for woven textile electrodes there is no electrolyte gel. Without electrolyte the contact pressure against the skin, and the moisture between skin and electrode become much more important [12,23,27]. A short period of time after application, a minuscule layer of perspiration accumulates between the epidermis and the conductive yarn [18,26]. In the current model, the only circuit term that is available to capture these physical characteristics is the $R_{s}$ electrolyte term. This indicates that in order to reduce impedance, some amount of contact pressure is required. Some recent literature has attempted to quantify pressure and moisture with the traditional model [23] while others propose modifications to the traditional model to better describe the sweat moisture accumulation and pressure dynamic of dry electrodes. One suggestion is to use a Constant Phase Element (CPE) instead of the capacitor term [28]. Another option is to add a third RC pair in series with the first two [29]. Now that this model is built and the fitting process exists in software, it is possible to more easily explore the relationships between these various factors and their corresponding circuit parameters.

Since the initial measurements were captured on stationary subjects, this analysis is only the first step in the long−term goal of modeling textile electrodes used for chronic applications. Note that in this application, since the objective is to characterize design properties, the goal was to remove and mitigate motion artifacts or individual differences. Further analysis on active subjects must be performed next. ECG data has been captured by Schauss using some of these same woven textile electrodes on subjects involved in non-seated activities [30].

### 4.1. Area Type

The first manufacturing parameter to discuss is the surface area of the electrode. The effect of surface area is consistent with that of the equation for a parallel plate capacitor and a volume resistance. The capacitance of a parallel plate capacitor is directly proportional to the area of the plates. The plates in this context are the conductive surface area of the electrodes. The resistance of a volume is inversely proportional to the surface area of that volume. The volume in this context is the conductive thread. The main observation from the area analysis is that a larger area patch, will have a higher capacitance and a lower resistance. As such, the smaller patches will collect lower amplitude waveforms through signal attenuation and capture more high frequency noise. The larger patches will collect larger amplitude waveforms, due to less signal attenuation and collect more low frequency noise. The insights about textile electrode area producing larger amplitude waveforms have been experimentally observed by others [4,8].

### 4.2. Yarn Type

For yarn type, the two yarns have the same yarn diameter, but different resistivities. The resistance of a volume is directly proportional to the resistivity of that volume. Resistivity does not appear in the equations for capacitance, and correspondingly we do not see a big trend in capacitance or phase with yarn type. The main observation from the yarn analysis is that electrodes with lower resistivity yarn will have a lower resistance, and a lower $Z_{E}$ impedance but relatively consistent capacitance. Lower resistivity means less signal attenuation for both the ECG waveform and the noise. This is confirmed by observing the relative spans of impedances between silver and steel yarns. Silver yarn has more varied impedance than steel, implying there is more noise impacting the mean ECG measurement from silver electrodes which is due to its lower resistivity. The insights about yarn type conductivity have been experimentally observed by others [8,13].

### 4.3. Pattern Type

The effect of pattern type on the electrode was smaller than the other two manufacturing parameters. The main impact of the pattern type is in its functional conductive surface area. All 8 electrodes in the pattern and yarn type comparison were manufactured with the same total area, 5.74 cm${}^{2}$. Since the electrodes were manufactured with conductive and non-conductive yarns, only a subset of the total surface area is conductive. A pattern which exposes less conductive yarn to the surface of the skin, will have a functionally smaller conductive surface area. As with the area swept electrodes, there is a downward trend in overall impedance as conductive area increases but because the conductive areas are comparatively closer together the trend less apparent.

## 5. Conclusions

This work applies the traditional double-time skin−electrode interface model to a variety of novel woven electrodes to provide the first relationship between manufacturing parameters and electrical impedances. This work is the first to compare simulated ECG waveforms with measured ECG waveforms collected from a set woven electrodes. The overall trends for the yarn type and surface area design parameters align with the expectations, indicating that the model captures the bulk behavior of the ECG system.

Several simplifying assumptions were made which future work should explore in more detail. Specifically electrode placement, surface pressure and skin moisture were out of scope. These are now thought to be significant factors for textile electrodes, so should be prioritized in future research with the computational model constructed in this work. Another area of future work would be to characterize the electro-chemical polarization of these specific woven electrodes similar to the work done previously on quantifying polarization for textile electrodes [31,32]. The work in this study may also be used to understand potential performance implications for alternative woven on-body textile electrodes. For example, a conformal, low-burden EEG system for hairless sites could be fabricated and worn, similar to a beanie or cap [33,34,35], or alternatively electro-dermal activity as monitored on the hand or wrist [36].

The main advantage of a computational model is its ability to quickly simulate and iterate designs. A validated model can provide the prospective opportunity to optimize the manufacture of woven textile electrodes. The model developed in this work brings that goal one step closer which ultimately enables ECG applications in wellness monitoring. In summary, this research facilitates a path for future development of these novel sensors to enable long−term ECG monitoring with woven textile electrodes.

## Figures and Tables

**Figure 1 sensors-23-00598-f001:**
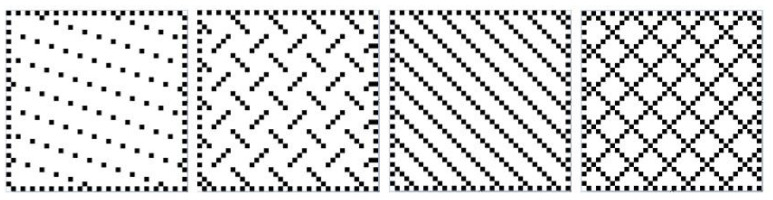
Pattern Types of woven electrodes. From Left to right: 1/15 Sateen, Broken Twill, Twill and Birdseye. Patterns are not to scale. Adapted from [21].

**Figure 2 sensors-23-00598-f002:**
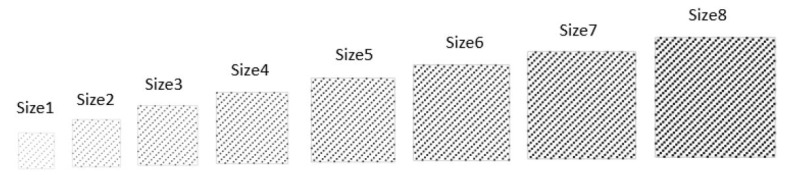
Relative Patch Surface Areas of woven electrode, smallest starts at 3.22 cm^2^ and increases up to 32.26 cm^2^. All are 1/15 Sateen pattern type. Areas not to scale. Adapted from [21].

**Figure 3 sensors-23-00598-f003:**
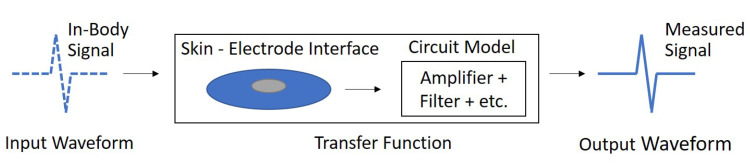
Notional Block Diagram of the ECG System.

**Figure 4 sensors-23-00598-f004:**
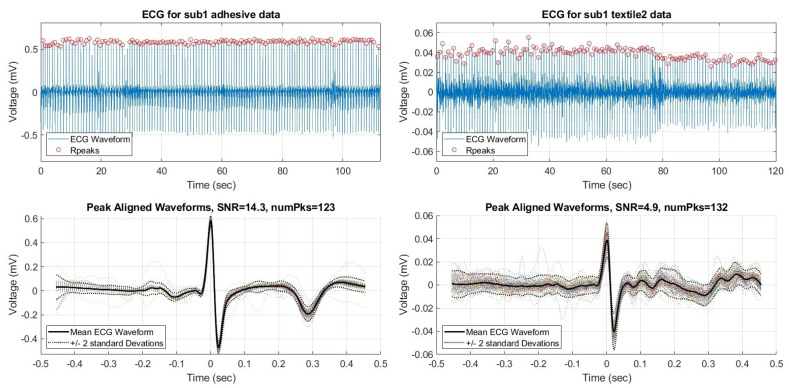
Time domain ECG data for subject 1 (**top**), collected using adhesive electrodes, and the corresponding Mean ECG waveform (**bottom**).

**Figure 5 sensors-23-00598-f005:**
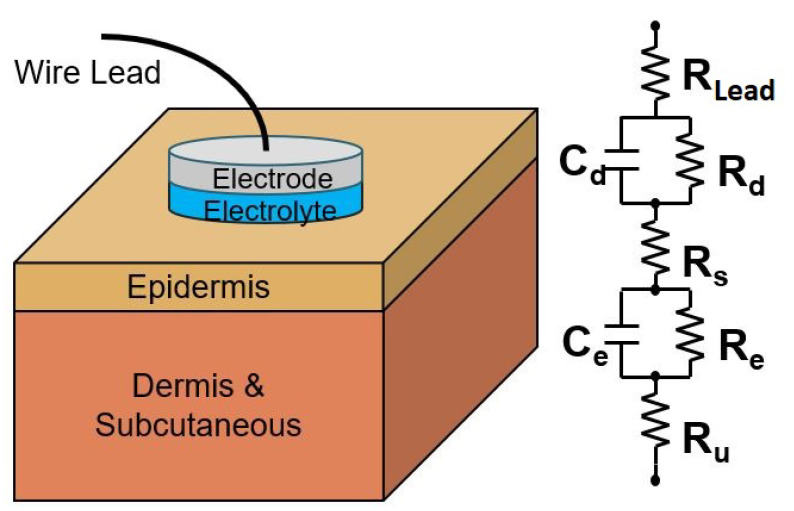
Traditional Double Time Model of the skin−electrode interface [9,14,18].

**Figure 6 sensors-23-00598-f006:**
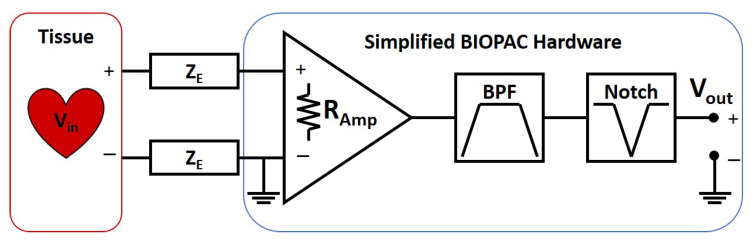
Simplified schematic diagram of the ECG system.

**Figure 7 sensors-23-00598-f007:**
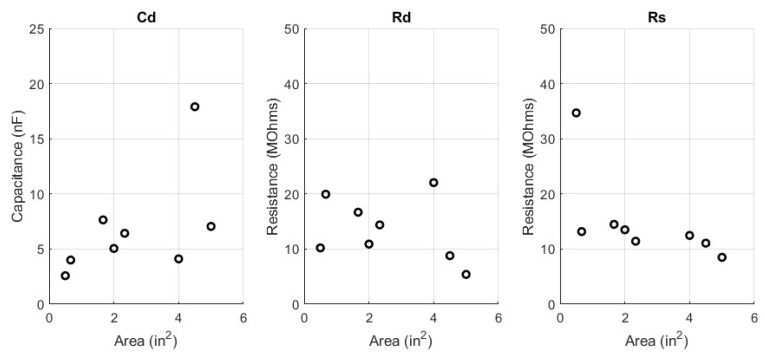
Surface Area Comparison versus Circuit Parameters; Yarn = Spun Steel, Pattern = 1/15 Sateen, $C_{d}$ (**left**), $R_{d}$ (**middle**), $R_{s}$ (**right**).

**Figure 8 sensors-23-00598-f008:**
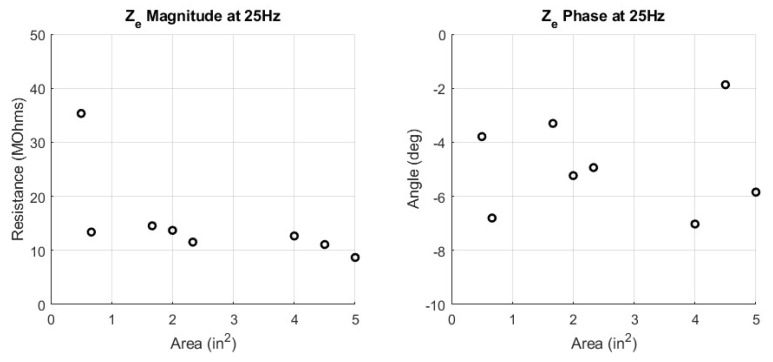
Surface Area Comparison versus Impedance $Z_{e}$; Yarn = Spun Steel, Pattern = 1/15 Sateen, Magnitude (**left**), Phase (**right**).

**Figure 9 sensors-23-00598-f009:**
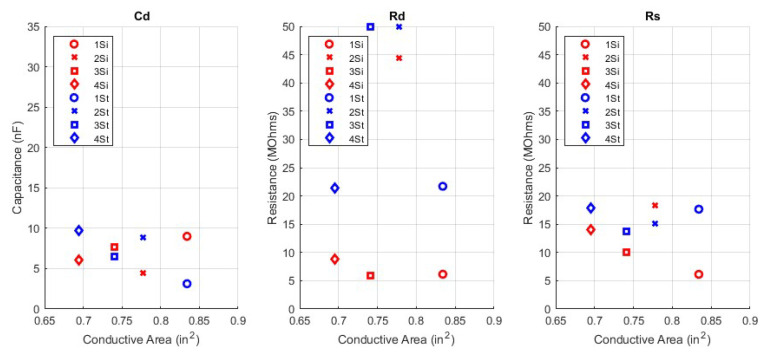
Pattern and Yarn type Comparison versus Circuit Parameters; Patch Surface Area = 5.74 cm${}^{2}$, $C_{d}$ (**left**), $R_{d}$ (**middle**), $R_{s}$ (**right**).

**Figure 10 sensors-23-00598-f010:**
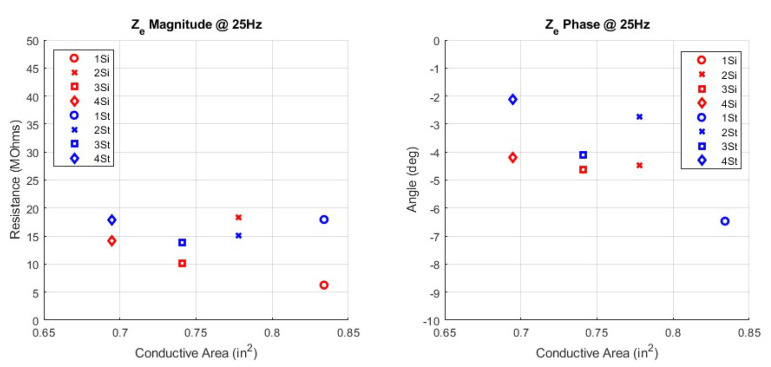
Pattern and Yarn type Comparison versus Impedance $Z_{e}$; Patch Surface Area = 5.74 cm${}^{2}$, Magnitude (**left**), Phase (**right**).

**Figure 11 sensors-23-00598-f011:**
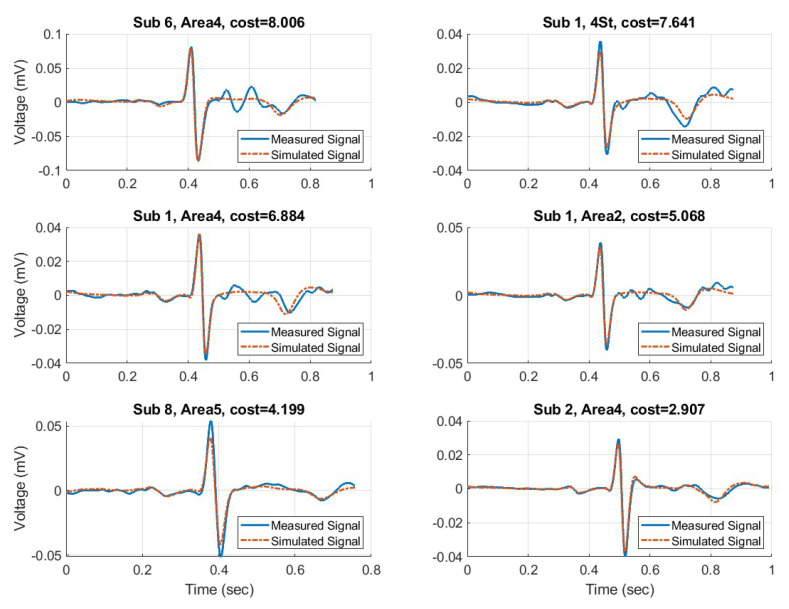
The 6 best fit simulated waveforms with their corresponding measured waveform.

**Figure 12 sensors-23-00598-f012:**
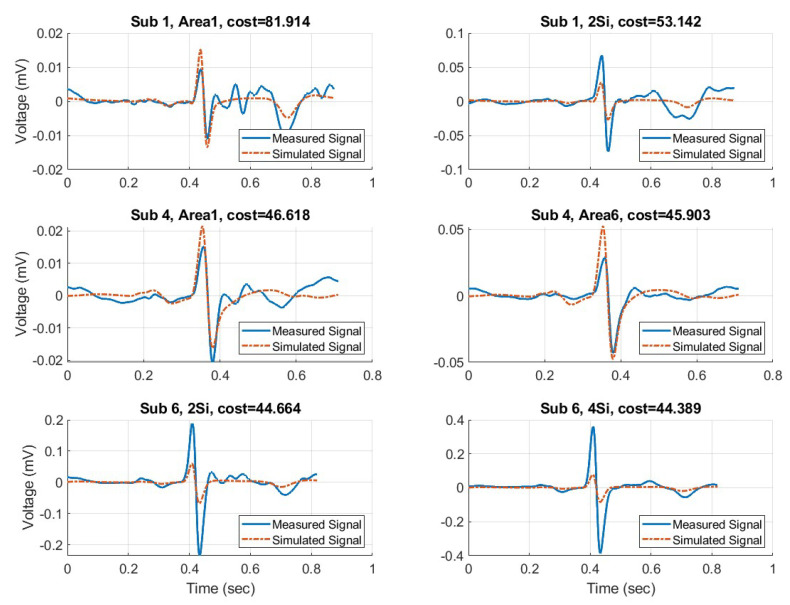
The 6 worst fit simulated waveforms with their corresponding measured waveform.

**Table 1 sensors-23-00598-t001:** Electrode manufacturing parameters.

ID	Label	Surface Area (cm)	Yarn Type	Pattern
1	Area1	1.27 × 2.54	Steel Spun	1/15 Sateen
2	Area2	1.27 × 3.38	Steel Spun	1/15 Sateen
3	Area3	2.54 × 4.22	Steel Spun	1/15 Sateen
4	Area4	2.54 × 5.08	Steel Spun	1/15 Sateen
5	Area5	2.54 × 5.92	Steel Spun	1/15 Sateen
6	Area6	3.81 × 6.76	Steel Spun	1/15 Sateen
7	Area7	3.81 × 7.62	Steel Spun	1/15 Sateen
8	Area8	3.81 × 8.46	Steel Spun	1/15 Sateen
9	1Si	1.68 × 3.38	Silver Nylon	1/15 Sateen
10	2Si	1.68 × 3.38	Silver Nylon	Broken Twill
11	3Si	1.68 × 3.38	Silver Nylon	Twill
12	4Si	1.68 × 3.38	Silver Nylon	Birds Eye
13	1St	1.68 × 3.38	Steel Spun	1/15 Sateen
14	2St	1.68 × 3.38	Steel Spun	Broken Twill
15	3St	1.68 × 3.38	Steel Spun	Twill
16	4St	1.68 × 3.38	Steel Spun	Birds Eye
17	Adh	* 19.61 cm^2^	* N/A	* N/A

* The clinical adhesive electrode used was a 5 cm diameter circular Ag/AgCl electrode. The circuit parameters for the adhesive electrode interface model are known from the work done by Assambo et al. [24]: Specifically *R*_*Lead*_ = 10 Ω, *C*_*e*_ = 0.9 uF, *R*_*e*_ = 35.2 Ωω, *R*_*u*_ = 2.6 kΩ.

## Data Availability

The MATLAB code containing the transfer function model, mean Waveform calculator, optimizer, and LOOCV, are available upon request but not yet posted publicly. The raw ECG data sets are available on Physionet.org. Please contact either author for more information.

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
