# Peer review of "Modeling the Design Characteristics of Woven Textile Electrodes for long−Term ECG Monitoring"

_sensors, 2023, doi:10.3390/s23020598_

Round 1

Reviewer 1 Report

This paper analyzes the use of new conductive textile electrodes as an alternative to adhesive gel electrodes. It seems that the main difficulty of textile electrodes is to understand the relationship between the manufacturing parameters and the ECG measurement. This paper addresses this challenge.

I suggest a revision based on the following points in order to improve the quality of the paper:

1.- I ask the authors to make an effort to list the main novelties and contributions of the paper with respect to the state-of-the-art. It is no clear if other authors are developing similar research.

2.- Equation (1) refers to the FFT, which is known it can be applied for stationary signals. Since it seems that the electrodes can monitor the current status of the human body in real daily conditions, it seems that (for example during efforts, physical work, etc.) the “stationary” condition is not always met. Please develop.

3.- It would be interesting for the readers to plot in a same graph simultaneously acquired ECGs with gel and textile electrodes.

4.- I miss a comparison of the ECG acquisitions made with the different prototype electrodes and a comparison with the reference gel electrode.

I hope this revision can help the authors to improve the quality and readability of the paper.

Author Response

Response to Reviewer #1: 
I ask the authors to make an effort to list the main novelties and contributions of the paper with respect to the state-of-the-art. It is no clear if other authors are developing similar research.  

The paper has been modified with several sentences in the abstract and introduction to explicitly state the novel contributions of the paper. This paper performs an analysis of a comprehensive sweep of novel woven electrode types to provide a novel understanding of which manufacturing design parameters most impact the electrical characteristics of the electrode. 

Equation (1) refers to the FFT, which is known it can be applied for stationary signals. Since it seems that the electrodes can monitor the current status of the human body in real daily conditions, it seems that (for example during efforts, physical work, etc.) the “stationary” condition is not always met. Please develop.  

A section was added to the discussion suggesting a future analysis be performed on a dataset of subjects who are not seated. Note that in this application, since the objective is to characterize design properties, the goal was to remove and mitigate motion artifact or individual differences.  

It would be interesting for the readers to plot in a same graph simultaneously acquired ECGs with gel and textile electrodes.  

A plot of an ECG signal captured with a textile electrode is provided alongside the plot of the ECG data captured with a gel electrode for the same subject to figure 4. Note the different y-axis.  

I miss a comparison of the ECG acquisitions made with the different prototype electrodes and a comparison with the reference gel electrode.   

The acquisition of the signals was not specifically a part of this work, as the data had been collected in a previous study and was compared in "Detection of the complete ECG waveform with woven textile electrodes” by Arquilla et al.. This is reference [21] in the paper. The two acquisitions were assumed to be identical except for the electrodes under test. The same acquisition process was performed in all data collection (same subjects, same circuitry etc.). In this work, the adhesive electrode set was used to define the “ideal in-body" signal and from there the remaining woven electrode sets were analyzed.  

Reviewer 2 Report

In the present manuscript the authors provide a relationship between the manufacturing parameters of seventeen different types of conductive woven textile electrodes for ECG measurements and their electrical characteristics. More in detail, the authors show how yarn type, pattern type and surface area are related to the skin-electrode impedance. Simulated ECG waveforms (obtained by means of the identified circuit parameters) have been compared with the corresponding measured waveforms.

The paper is clearly written, data and analyses presented appropriately, and the discussion of the results is convincing.

In conclusion, I believe that the results presented in the manuscript will be of significant interest for the community and I recommend publication.

I would suggest to the authors, however, to clarify/correct the following minor points:

- in paragraph 2.1, it could be useful to provide more details about the procedure of removal of artifacts from ECG signals;

- in line 114, the name of the coefficients should be Ai (i = 0, 1, … n) and Bi (i = 0, 1, … m), instead of “AB”;

- in line 146, the name of the procedure should be “Leave-One-Out Cross-Validation” (as correctly reported in the following);

- in line 160 the comma after Cd should be removed

- in figure 5, the resistance of the lead wire should be indicated as RLead (instead of Rlead)

Author Response

Response to Reviewer #2 

In paragraph 2.1, it could be useful to provide more details about the procedure of removal of artifacts from ECG signals;  

 Additional details about peak identification, and outlier identification are provided on lines 126 through 129.  

In line 114, the name of the coefficients should be Ai (i = 0, 1, … n) and Bi (i = 0, 1, … m), instead of “AB”;  

The reviewer is correct. “AB” has been removed and the coefficients changed to align with your suggestion.  

In line 146, the name of the procedure should be “Leave-One-Out Cross-Validation” (as correctly reported in the following);  

These grammatical errors have been corrected and dashes are now added to the procedure name.  

In line 160 the comma after Cd should be removed  

The comma was removed after Cd

In figure 5, the resistance of the lead wire should be indicated as RLead (instead of Rlead 

The “L” in Lead in Figure 5 is now capitalized.  

Reviewer 3 Report

In this study, the authors modeled the design characteristics of woven textile electrodes for long-term ECG monitoring. This study showed how yarn type, pattern type, and surface area are related to the skin-electrode impedance and correspondingly the ECG waveform. This relationship provides new insight into the design of textile electrodes and informs new areas of exploration for even better designs. Although being interesting, I find that there are some major issues with the paper that require addressing prior to this being considered for publication in this journal. I have identified the main points for consideration below:

1.     This manuscript has some spelling typos, style errors and grammatical errors. Please carefully check the whole manuscript.

2.     The electrode-skin impedance is quite different for various subjects. How did the authors address the subject-dependent impedance issues in this study?

3.     In addition to the electrode-skin impedance, the non-polarization performance of the electrode is very critical for smooth baseline. How did yarn type, pattern type, and surface area affect the nonpolarizable properties?

4.     How many subjects are involved in this study?

5.     In the discussion section, I suggest the author discussion the feasible of textile electrodes for EEG recording at hairless sites. In addition, some related references are recommended to be cited in the introduction section, such as Journal of Neural Engineering 18 (2021) 046016; Journal of Neural Engineering 17 (2020) 051004; Sensors 2022, 22(21), 8510; Adv. Mater. Technol. 2022, 7, 2100612.

Author Response

Response to Reviewer #3 

This manuscript has some spelling typos, style errors and grammatical errors. Please carefully check the whole manuscript.  

The whole manuscript was thoroughly reviewed for style and grammar errors. 

The electrode-skin impedance is quite different for various subjects. How did the authors address the subject-dependent impedance issues in this study?  

A normalized cost function was used to obtain a set parameter which represents the best fit across all subjects. The normalized cost function indicated in Equation 5 computes the cumulative costs of all subjects added together. Several sentences were added for clarity to explain this step in the analysis process.  

In addition to the electrode-skin impedance, the non-polarization performance of the electrode is very critical for smooth baseline. How did yarn type, pattern type, and surface area affect the nonpolarizable properties?  

The relationship between manufacturing parameters and electro-chemical polarization was not characterized in this work. This topic has been added as an area of future research in the conclusion statement.  

How many subjects are involved in this study?  

Six human subjects were involved in this study. This was stated originally on line 81. This is now on line 86 after implementing reviewer feedback.  

In the discussion section, I suggest the author discussion the feasible of textile electrodes for EEG recording at hairless sites. In addition, some related references are recommended to be cited in the introduction section, such as Journal of Neural Engineering 18 (2021) 046016; Journal of Neural Engineering 17 (2020) 051004; Sensors 2022, 22(21), 8510; Adv. Mater. Technol. 2022, 7, 2100612. 

The work in this study may be used to understand potential performance implications for alternative woven on-body textile electrodes.  Several examples were added to the discussion section. Several of the suggested references were added to the paper.  

Round 2

Reviewer 1 Report

The authors have replied my questions

Reviewer 3 Report

The authors have well addressed all my comments. Thanks for an interesting paper.